# Analysis of Saliva Lipids in Breast and Prostate Cancer by IR Spectroscopy

**DOI:** 10.3390/diagnostics11081325

**Published:** 2021-07-23

**Authors:** Lyudmila V. Bel’skaya, Elena A. Sarf, Victor K. Kosenok

**Affiliations:** 1Biochemistry Research Laboratory, Omsk State Pedagogical University, 644099 Omsk, Russia; belskaya@omgpu.ru; 2Department of Oncology, Omsk State Medical University, 644099 Omsk, Russia; victorkosenok@gmail.com

**Keywords:** saliva, infrared spectroscopy, lipids, breast cancer, prostate cancer

## Abstract

We have developed a method for studying the lipid profile of saliva, combining preliminary extraction and IR spectroscopic detection. The case–control study involved patients with a histologically verified diagnosis of breast and prostate cancer and healthy volunteers. The comparison group included patients with non-malignant pathologies of the breast (fibroadenomas) and prostate gland (prostatic intraepithelial neoplasia). Saliva was used as a material for biochemical studies. It has been shown that the lipid profile of saliva depends on gender, and for males it also depends on the age group. In cancer pathologies, the lipid profile changes significantly and also depends on gender and age characteristics. The ratio of 1458/1396 cm^−1^ for both breast and prostate cancer has a potential diagnostic value. In both cases, this ratio decreases compared to healthy controls. For prostate cancer, the ratio of 2923/2957 cm^−1^ is also potentially informative, which grows against the background of prostate pathologies. It is noted that, in all cases, changes in the proposed ratios are more pronounced in the early stages of diseases, which increases the relevance of their study in biomedical applications.

## 1. Introduction

Recently, the attention of researchers to the study of the properties of human saliva as a material with unique properties and diagnostic capabilities has been increasing [1,2,3,4,5]. A promising direction is the use of various spectroscopic methods, including infrared (IR) spectroscopy, to study the composition of saliva [6,7,8,9,10]. However, studies of saliva by IR spectroscopy are often limited to its use for the diagnosis of the oral cavity diseases [11,12,13,14,15]; nevertheless, the diagnostic potential of saliva is much wider [16,17,18,19,20,21]. It is important to note that there is an increasing interest in saliva research for cancer diagnosis [22,23,24,25].

Studies by a number of authors show the important role of lipids in the processes of carcinogenesis and suggest using these indicators in the diagnosis of cancer [26,27]. However, the determination of lipids in saliva is associated with a number of difficulties, since their content is low. Modification and increase in the accuracy of lipid determination in most methods is achieved by preliminary extraction and obtaining a transparent solution of lipids for their subsequent enzymatic evaluation [28,29,30]. The most common method of lipid extraction is the Folch method, which allows 90–95% of all cellular lipids to be extracted [31,32]. The disadvantage of this method is the fact that solvent mixtures containing alcohol also extract non-lipid substances (sugars, amino acids, salts, etc.). To remove non-lipid impurities, the lipid extract must be washed with water or weak saline solutions. However, this leads to a partial loss of acidic lipids, and in some cases also contributes to the formation of very stable emulsions. Purification from non-lipid impurities can be avoided by using IR spectroscopy to analyze the obtained extract [33], since absorption bands corresponding to vibrations of lipid functional groups without preliminary sample treatment are not always informative due to the overlap with water absorption bands [34,35,36].

We have developed a method for the quantitative determination of lipids in saliva, in which, after extraction with a chloroform/ethanol mixture, lipids are determined by IR spectroscopy [37,38]. The bands corresponding to the stretching and deformation vibrations of the methyl and methylene groups of lipids and fatty acids were selected as analytical absorption bands: 1396, 1458, 2853, and 2923 cm^−1^. These bands do not intersect with the absorption bands of proteins and nucleic acids, thus avoiding the preliminary cleaning stage of lipids from non-lipid impurities. A multifactor regression model was constructed, which allows experimental data to be described with an error not exceeding 12%. On the example of ovarian and endometrial cancer, it was shown that in cancer, the lipid content in saliva decreases and a ratio of the intensities of absorption bands 2923/2957 cm^−1^ was proposed, the decrease in which is statistically significant both at the initial and at the advanced stages of the disease and can be used as a potential diagnostic criterion [38]. We also showed earlier that the composition of saliva has pronounced sex differences [39], which are also manifested in IR spectra [40]. In the framework of this study, we analyzed the lipid profile of saliva in healthy volunteers of both sexes and established reference lipid values for men and women. After that, typical male and female cancers (breast cancer and prostate cancer) were selected. Using the example of these groups, a comparison of lipid profiles in cancer and benign pathologies was carried out to determine their potential diagnostic value.

## 2. Materials and Methods

### 2.1. Study Design

The case–control study involved volunteers who were divided into three groups: the main group, the comparison group and the control group. The main group included patients with histologically verified diagnosis of cancer, including breast cancer (ductal and lobular carcinoma, *n* = 30) and prostate cancer (acinar adenocarcinoma, *n* = 21). The comparison group included patients with non-malignant pathologies of the breast (fibroadenomas, *n* = 47) and prostate (prostatic intraepithelial neoplasia PIN I-II, *n* = 21). A detailed description of the studied groups is given in Table 1 and Table 2.

The control group consisted of 100 healthy volunteers in whom, as part of a routine medical examination, no breast and prostate pathologies were revealed (58 males and 42 females). At the time of the study, all patients had no signs of inflammatory processes, diseases of the oral cavity and periodontal disease. All volunteers had not smoked for at least three years, had no clinically significant concomitant diseases, including diabetes mellitus, cardiovascular and autoimmune diseases. Saliva was used as a material for biochemical studies.

### 2.2. Saliva Collection and Storage Technique

Saliva samples were collected in the morning on an empty stomach at 8–10 a.m. (time of maximum secretion) in accordance with circadian rhythms [41]. Before collection, volunteers rinsed their mouths with distilled water, after which saliva was collected by spitting into sterile polypropylene tubes. The volume of saliva for analysis was 1 mL. The unstimulated whole saliva samples were centrifuged (10,000× *g* for 10 min) to remove cellular debris and to minimize the turbidity of saliva, which could negatively impact the accuracy of analysis [42]. Extraction and IR spectroscopic examination were performed immediately after centrifugation (without freezing and storage).

### 2.3. Saliva Analysis by IR Spectroscopy

Lipids from saliva samples were extracted using Folch solution (chloroform:ethanol = 2:1, vol.) [37,38]. When analyzing biological material, 200 μL of saliva was diluted with 800 μL of 0.9% NaCl, then the samples were extracted twice with 2 mL of Folch solution. The combined organic phase was settled for 24 h, and then centrifuged (10,000× *g* for 10 min) for a more complete phase separation. The upper layer was carefully decanted and the bottom layer was selected for IR spectroscopy. Extracts with a volume of 50 μL were dried for 30 min on a substrate of zinc selenide in a thermostat at 37 °C. The infrared absorption spectra were registered using an FT-801 Fourier IR spectrometer (Simex, St. Petersburg, Russia) in the range of 500–4000 cm^–1^. Spectra were recorded with a scan number of 32 with a resolution of 4 cm^−1^. Three spectra were collated per sample/patient. The results were presented as an averaged (or levelled) spectrum.

ZaIR 3.5 software (Simex) was used to carry out baseline correction and normalization of FTIR spectra. A background (air) measurement was taken for every sample processed. The peaks corresponding to CO_2_ vibrations were removed using the “straight line generation” option in the ZaIR 3.5 software (Simex). Raw spectra were pre-processed using a simple two-point linear subtraction baseline correction method. Two points, 900 and 1850 cm^−1^, were selected outside the wavenumber region of interest that showed no variation across all samples. Spectra were then vector normalized. Spectrum smoothing was not performed.

Previously, we simulated solutions with different lipid content and established absorption bands in the IR spectrum, the characteristics of which correlate with the level of lipids [37]. As analytical absorption bands, we selected the bands corresponding to the stretching (ν_s_—symmetric, ν_as_—asymmetric) and deformation (δ) vibrations of the methyl and methylene groups of lipids and fatty acids: 1396 cm^−1^ (δCH_3_), 1458 cm^−1^ (δCH_2_), 2853 cm^−1^ (ν_s_CH_3_), 2923 cm^−1^ (ν_as_CH_2_) and 2957 cm^−1^ (ν_as_CH_3_) [37,38]. In parallel, we measured the intensity (H) and area (S) of the corresponding peaks in the IR spectra of the samples (Figure 1). Additionally, the ratios of the intensities and areas of the absorption bands at 2923/2957 and 1458/1396 cm^−1^ were calculated.

### 2.4. Statistical Analysis

Statistical analysis of the data obtained was performed using the Statistica 10.0 (StatSoft, Tulsa, OK, USA) program by a nonparametric method using the Mann–Whitney U-test for pairwise comparison and the Kruskal–Wallis test for comparing three or more groups.

Multivariate comparison was carried out by the method of Discriminant analysis (Statistica 10.0, StatSoft). Discriminant analysis is used to decide which variables distinguish (discriminate) between two or more groups. The basic idea of discriminant analysis is to determine if groups differ in the mean of a variable (or a linear combination of variables), and then use that variable to predict for new members whether they belong to a particular group. If there are more than two groups, then more than one discriminant function can be evaluated. For example, when there are three groups, we can evaluate: (1) a function for discrimination between group 1 and groups 2 and 3 taken together (Root 1), and (2) another function for discrimination between group 2 and group 3 (Root 2). On a scatterplot of canonical values, we can visually assess the position of the groups relative to the vertical (Root 1) and horizontal axis “0-0” (Root 2). Groups that are statistically significantly different from others are shown in the diagrams with ellipses of the corresponding color.

Percentages of changes in the intensity of peaks attributable to cancer and non-malignant pathologies to control peaks are calculated from the average values for the respective groups.

## 3. Results

### 3.1. Analysis of the Salivary Lipid Profile of the Control Group

At the first stage of the study, we tested the hypothesis that the lipid profile of saliva significantly depends on the gender and age of the volunteers. It was shown that the intensities and areas of the absorption bands at 1396, 1458 and 2957 cm^−1^, as well as the ratios of 2923/2957 and 1458/1396 cm^−1^, are statistically significantly different (Table 3).

Since the intensity and area of absorption bands change unidirectionally, in further calculations we used only one characteristic of the IR spectra—intensity. The additional consideration of volunteer age made it possible to establish that in the group of women, age does not significantly affect the characteristics of the IR spectra, while for the males group this factor is important (Figure 2 and Figure 3). The scatter diagram of canonical values shows that the vertical axis “0-0” separates the groups of males over 50 (to the right of the axis) and females, while the horizontal axis separates males of different ages (Figure 2). It was found that the intensity of the absorption band at 1458 cm^−1^ increases with age (Figure 3b) and the intensities of the absorption bands at 2923 and 2957 cm^−1^ decrease (Figure 3e,f). In general, both for males and females, it is shown that the ratio of 2923/2957 cm^−1^ decreases with age and the ratio of 1458/1396 cm^−1^ increases (Figure 3g,h).

Due to the fact that the lipid profile has a pronounced dependence on gender, at the next stage of the study, typical female and male diseases, namely breast and prostate cancer, were selected.

### 3.2. Saliva Lipid Profile in Breast Cancer

Since there were no differences between the age groups for females in the lipid profile of saliva, the comparison was made with the control group without additional consideration of age (Table 3, Females). Nevertheless, it was shown that there are no statistically significant differences in age between the study groups (Table 4).

It has been shown that in the case of pathologies of the mammary glands, both benign and malignant, changes in the lipid profile are observed (Table 4). Changes affect almost all absorption bands in the IR spectra of saliva with the exception of the 2853 cm^−1^ band (Table 4). When compared with the control group, it can be seen that for some of the absorption bands, the changes are unidirectional, for example, the absorption bands at 1396, 1458 and 2957 cm^−1^ (Figure 4a). For the absorption band at 2923 cm^−1^, the intensity changes in different directions: it increases in breast cancer and decreases in fibroadenomas (Figure 4a). However, statistically significant differences are observed only for the absorption bands at 1396 and 1458 cm^−1^. Nevertheless, between the group of patients with breast cancer and fibroadenomas, significant differences were found for the intensities of absorption bands at 1458, 2853, 2923, and 2957 cm^−1^ (Figure 4b).

Taking into account the stage of the disease, it was shown that both the early and widespread stages of the disease in terms of the lipid profile differ from the healthy control (Figure 5), however, no significant differences between the stages were revealed (Figure 5 and Figure 6). Nevertheless, it was preliminary shown that changes in the intensities of lipid absorption bands are more pronounced for advanced stages of breast cancer (Figure 6). Due to the small sample size, the revealed differences were not statistically confirmed, except for the ratio 1458/1396 cm^−1^, the change in intensity of which is more pronounced in the early stages.

A comparison of the previously obtained data on the lipid profile of saliva in ovarian and endometrial cancer [38] shows that in the case of gynecological pathologies, the changes relate to a greater extent to the absorption bands at 2853, 2923, and 2957 cm^−1^ (Figure 7), the change in the ratio of 2923/2957 cm^−1^. In the case of breast cancer, the absorption bands at 1396 and 1458 cm^−1^ and, accordingly, the ratio of 1458/1396 cm^−1^, which can potentially be used as diagnostic criteria, change to a maximum.

### 3.3. Saliva Lipid Profile in Prostate Cancer

In the case of pathologies of the prostate gland, it becomes necessary to use a control group of the appropriate age, since for males age-related differences are most clearly manifested in the salivary IR spectra. Therefore, a control group over 50 years old was selected (Table 5). 

It was found that both for malignant and non-malignant pathologies of the prostate gland, the intensity and area of the absorption bands of lipids in saliva decreases (Figure 8a), while the ratio of 2923/2957 cm^−1^ significantly increases, and 1458/1396 cm^−1^ significantly decreases (Figure 8a). A comparison of the lipid profile in prostate cancer and prostatic intraepithelial neoplasia showed that the characteristics of the absorption bands at 1396 and 1458 cm^−1^ change in different directions, which was not previously noted for breast pathologies (Figure 8b).

A comparison of the lipid profile at different Gleason grades showed that with a lower Gleason grade, the changes are more pronounced, which can be used for the timely diagnosis of prostate pathologies (Figure 9).

## 4. Discussion

Lipids are the main components of the cell membrane required for a variety of biological functions, including cell growth and division to maintain cell integrity [43,44]. Cholesterol is a precursor of bile acids and steroid hormones; it can cause an increase in tumor angiogenesis, a decrease in tumor apoptosis, and an increase in tumor cell proliferation [45]. One of the proposed mechanisms involves the vital role of cholesterol in cell membranes, which can influence various signaling pathways [46]. It is known that increased synthesis of fatty acids is one of the most important abnormalities in the metabolism of cancer cells and is necessary for both carcinogenesis and the survival of cancer cells [26,47,48]. To maintain cell proliferation in cancer, it is necessary to enhance lipid metabolism [27,49]; therefore, the lipid level decreases in the group of patients with malignant tumors [50]. Certain atypical tendencies are observed in the lipid profiles of blood plasma of cancer patients [51,52,53,54,55,56,57,58]. The key role of lipid metabolism in metastatic colonization has also been shown [59].

We have proposed a method for preliminary lipid extraction, which allows us to focus exclusively on the lipid component of the IR spectrum of saliva without interfering with the absorption bands of proteins and nucleic acids [37,38]. It has been shown that, after extraction, the intensities of lipid absorption bands increase significantly, which in turn ensures greater accuracy in determining the characteristics of spectra (intensity and peak area) and can provide valuable information about changes in the structure of lipids that appear against the background of various pathologies of the human body. In particular, it is known that the 3050–2800 cm^−1^ interval, which contains vibrations of methyl and methylene groups of saturated and unsaturated alkyl chains, can be useful for assessing the permeability of cell membranes, as well as the processes of oxidative modification of proteins [60]. A potentially informative intensity ratio of 2923/2957 cm^−1^, which shows the ratio of unbranched and branched molecules of lipids and fatty acids (CH_2_/CH_3_). For example, against the background of oncological pathology, this ratio increases in comparison with the control [61], which indicates more branched chains and / or shorter chains of lipids and fatty acids in this case compared to the norm. We have shown for the first time on the example of saliva that the interval of 1300–1500 cm^−1^ is also informative for studying the lipid profile.

Despite the confirmed important role of lipids in the processes of carcinogenesis, their content even normally has its own characteristics, which should be taken into account when planning the study design and forming groups. At the same time, the nature of the change in the lipid profile in different oncological diseases is also significantly different, which is apparently associated with the peculiarities of a particular pathology. Thus, in breast cancer, the intensities of absorption bands at 1396 and 1458 cm^−1^ change significantly, which distinguishes this pathology from ovarian and endometrial cancer and is consistent with the data of Izabella Ferreira that the absorption in the range of 1433–1302.9 cm^−1^ in saliva is higher than for control and non-malignant breast pathologies [23]. For prostate cancer, the ratio of 2923/2957 cm^−1^ changes significantly. Since different pathologies change the ratios corresponding to either stretching (ν_as_CH_2_/ν_as_CH_3_) or bending vibrations (δCH_2_/δCH_3_) of methylene and methyl groups in the structure of lipids, this can serve as an indicator of more subtle differences in the structure of lipids.

According to the known data, changes in the proportion of certain lipid classes during carcinogenesis are very variable [62,63,64,65]. Thus, a typical feature is an increase in the ratio of free cholesterol to phospholipids. This can occur due to a decrease in the proportion of cholesterol esters [62] or a decrease in the total content of phospholipids (that is, the ratio of phospholipids to protein decreases) [63]. As a rule, the proportion of polyunsaturated fatty acid residues in lipids decreases, the proportion of monounsaturated fatty acids may slightly increase, and the ordering increases as a result of a higher proportion of cholesterol [64]. One of the reasons for changes in the lipid composition during tumor transformation is also oxidation induced by reactive oxygen species (ROS). At the initial stages of tumor formation, the amount of ROS in the transforming cells increases sharply and the deactivation systems cannot cope with them [65]. Lipids, nucleic acids and proteins begin to be damaged. The sensitivity of lipids to oxidation decreases with a decrease in the number of double bonds in the fatty acid; cholesterol is the most difficult to oxidize. The first target is the remains of arachidonic and linoleic acids in lipids. At the next stage, the formation of polymers with ether, peroxy and carbon–carbon bonds, intramolecular rearrangements giving various cyclic, keto and epoxy derivatives is possible. However, the most dangerous products are aldehydes resulting from the decomposition of hydroperoxide [65]. Most often, the breaking of the carbon–carbon bond near the -OOH group occurs by the β-cleavage mechanism. The longer the aldehyde detached from the methyl end of the fatty acid, the more cytotoxic it has. Aldehyde products form covalent adducts with lipid and protein molecules, disrupting their functioning. All these changes can be reflected in the IR spectra of saliva; however, the preliminary extraction and concentration of lipids is required to obtain more complete information.

## 5. Conclusions

A method for studying the lipid profile of saliva has been developed, which combines preliminary extraction by the modified Folch method and IR spectroscopic detection. It has been shown that the lipid profile depends on gender, and for males also on the age group. In cancer pathologies, the lipid profile changes significantly and also depends on gender and age characteristics. For breast cancer, the intensities of lipid absorption bands increase, while for prostate cancer they decrease. The ratio of 1458/1396 cm^−1^ for both breast and prostate cancer has a potential diagnostic value. In both cases, this ratio decreases compared to healthy controls. For prostate cancer, the ratio of 2923/2957 cm^−1^ is also potentially informative, which grows against the background of prostate pathologies. It is noted that in all cases, changes in the proposed ratios are more pronounced in the early stages of diseases, which increases the relevance of their study in biomedical applications.

The limitations of the study include the small sample size, which does not allow a more in-depth analysis of the effect of the stage of the disease to be made, as well as the molecular subtype in the case of breast cancer. For a deeper understanding of the ongoing changes, it is planned to conduct a parallel chromatographic analysis of lipids and a correlation analysis of the results obtained by two methods. In the continuation of the study, it is also planned to expand the groups and introduce additional types of oncological diseases that occur in both males and females (lung cancer and thyroid cancer) to confirm the hypotheses formulated in this work.

## Figures and Tables

**Figure 1 diagnostics-11-01325-f001:**
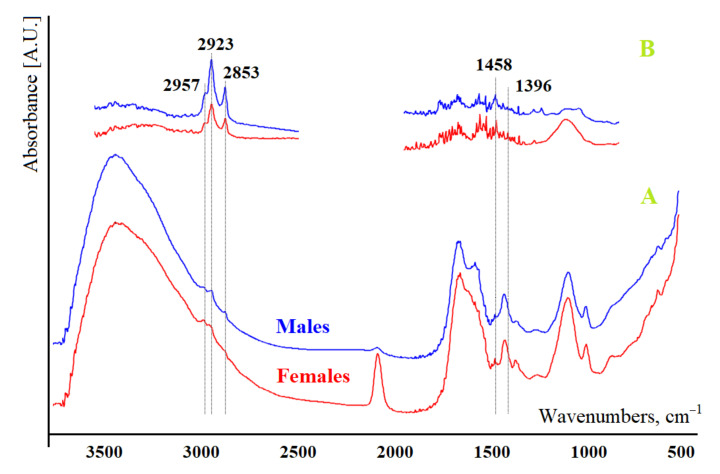
An example of the IR spectrum of the saliva of the control group before extraction (**A**) and after extraction (**B**).

**Figure 2 diagnostics-11-01325-f002:**
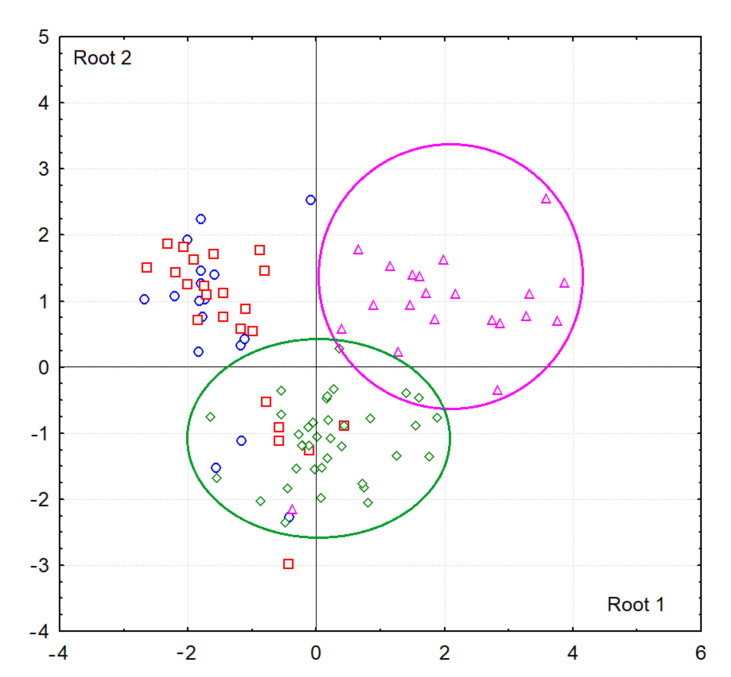
Scatter plot of canonical values for the control group depending on gender and age (blue circles—females under 50, red squares—females over 50, green diamonds—males under 50, pink triangles—males over 50). The ellipses mark the groups, the differences of which with others are statistically significant.

**Figure 3 diagnostics-11-01325-f003:**
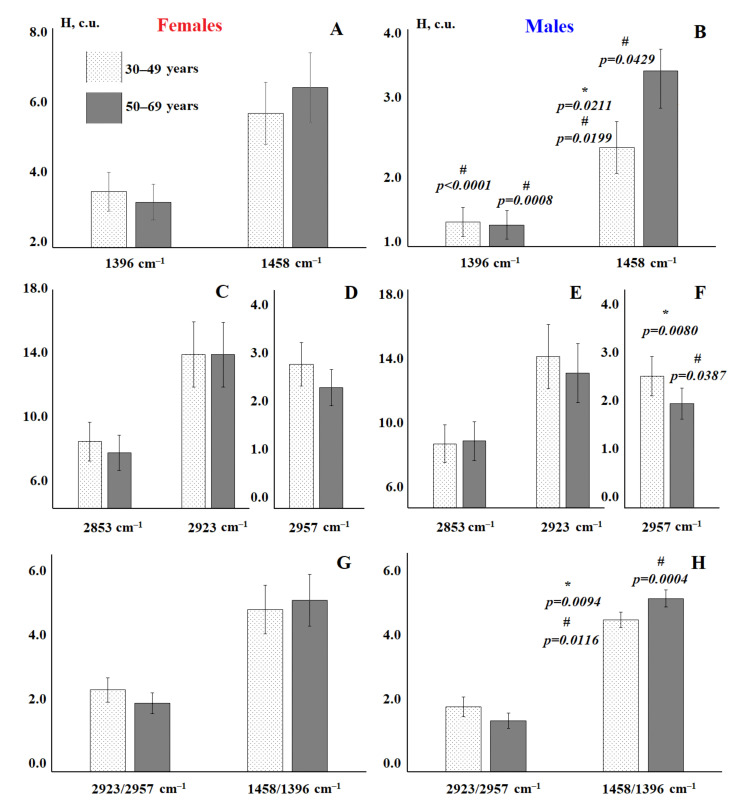
Intensity of lipid absorption bands in the IR spectra of saliva of males and females of the control group depending on age: absorption bands at 1396 and 1458 cm^−1^ (**A**,**B**), 2583 and 2923 cm^−1^ (**C**,**E**), 2957 cm^−1^ (**D**,**F**), ratios 2923/2957 and 1458/1396 cm^−1^ (**G**,**H**). *—differences between different age groups of the same gender are statistically significant, #—differences between volunteers of different gender within one age group are statistically significant, *p* ˂ 0.05. *p*-values are calculated using the paired Mann–Whitney test.

**Figure 4 diagnostics-11-01325-f004:**
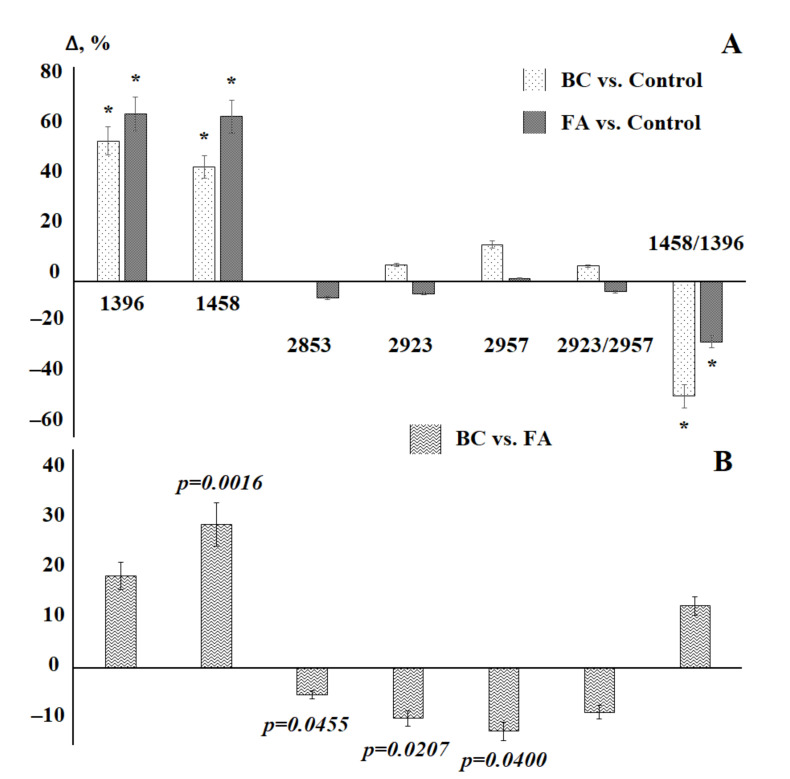
Changes in the intensity of absorption bands on the IR spectra of saliva in patients with breast cancer (BC) and fibroadenomas (FA) compared with the control group (**A**) and patients with breast cancer compared with fibroadenomas (**B**). *—differences are statistically significant, *p* < 0.05. *p*-values are calculated using the paired Mann–Whitney test.

**Figure 5 diagnostics-11-01325-f005:**
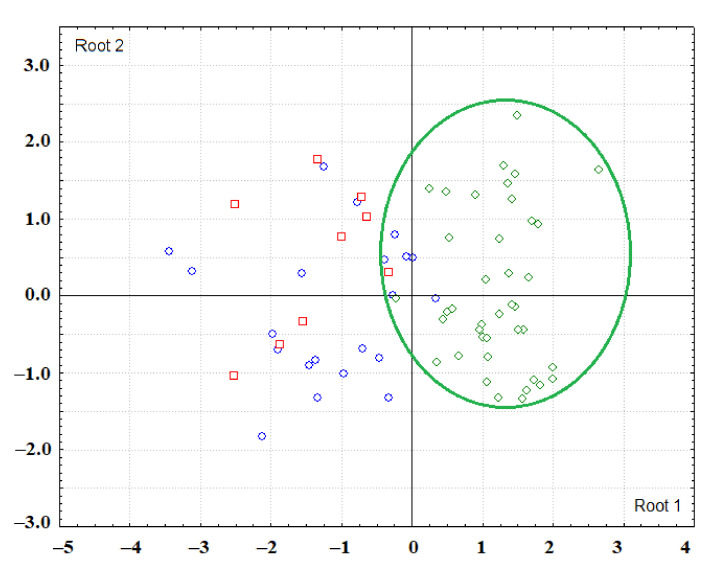
Scatter plot of canonical values for breast cancer patients by stage (blue circles—T_1-2_N_0-1_M_0_, red squares—T_3-4_N_0-2_M_0_) and fibroadenomas (green diamonds). The ellipses mark the groups, the differences of which with others are statistically significant.

**Figure 6 diagnostics-11-01325-f006:**
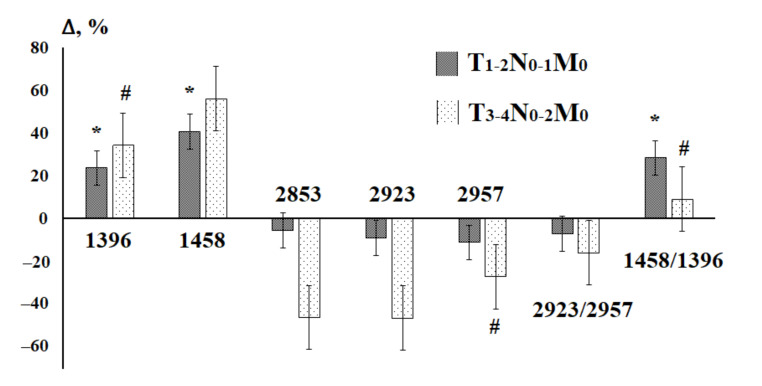
Relative change in the intensity of lipid absorption bands in the IR spectra of saliva in breast cancer patients, depending on the stage of the disease. *—differences with the control group are significant for T_1-2_N_0-1_M_0_, #—differences with the control group are significant for T_3-4_N_0-2_M_0_.

**Figure 7 diagnostics-11-01325-f007:**
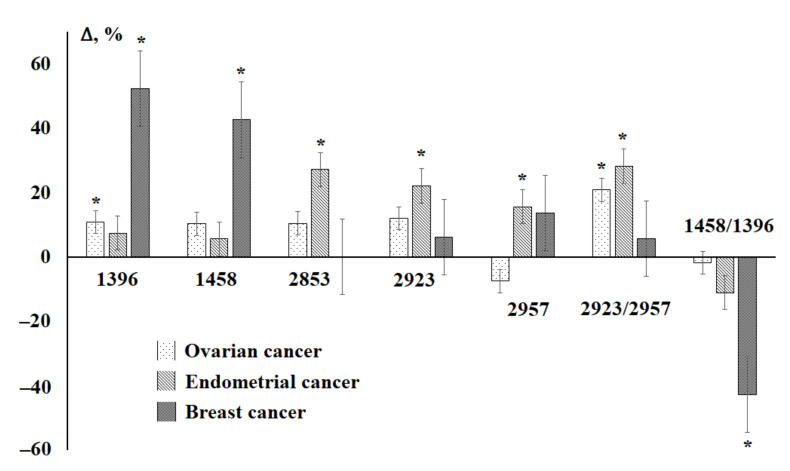
Relative changes in the intensities of the absorption bands of the salivary IR spectra in ovarian, endometrial and breast cancer. *—differences in comparison with the control group are statistically significant, *p* < 0.05.

**Figure 8 diagnostics-11-01325-f008:**
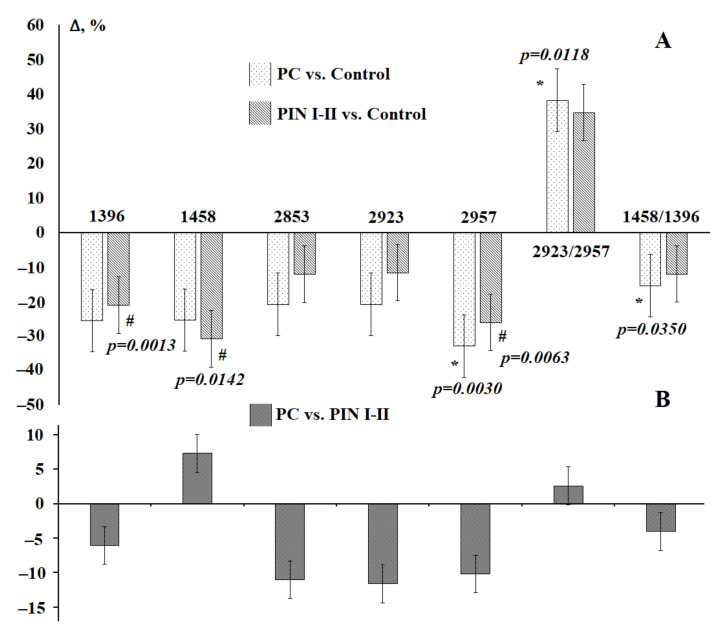
Changes in the intensity of the absorption bands of lipids in saliva with pathologies of the prostate gland: (**A**)—compared with the control group, (**B**)—compared with non-malignant pathologies. *p*-values are calculated using the paired Mann–Whitney test. */#—differences are statistically significant, *p* < 0.05.

**Figure 9 diagnostics-11-01325-f009:**
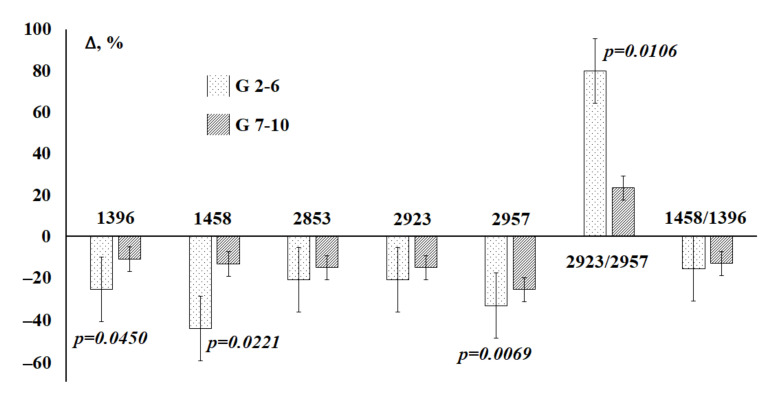
Changes in the intensity of lipid absorption bands in saliva in prostate cancer depending on Gleason grade compared to healthy controls. *p*-values are calculated using the paired Mann–Whitney test.

**Table 1 diagnostics-11-01325-t001:** Description of the group of patients with breast pathologies.

Group	Subgroup	Number of Patients (%)
Breast cancer		30
Stage	T_1-2_N_0-1_M_0_	20 (66.7%)
	T_3-4_N_0-2_M_0_	10 (33.3%)
Histological type	Ductal carcinoma	18 (60.0%)
	Lobular carcinoma	12 (40.0%)
The degree of anaplasia	G1	8 (26.7%)
	G2	12 (40.0%)
	G3	10 (33.3%)
Molecular subtypes	Luminal A	3 (10.0%)
	Luminal B	20 (66.7%)
	HER2+	7 (23.3%)
Fibroadenomas		47

**Table 2 diagnostics-11-01325-t002:** Description of the group of patients with prostate pathologies.

Group	Subgroup	Number of Patients (%)
Prostate cancer		21
Stage	T_2_N_0-1_M_0_	16 (76.2%)
	T_3_N_0-1_M_0_	5 (23.8%)
Gleason grade	G 2–6	9 (42.9%)
	G 7–10	12 (57.1%)
Prostatic intraepithelial neoplasia		21
PIN	PIN I	10 (47.6%)
	PIN II	11 (52.4%)

**Table 3 diagnostics-11-01325-t003:** Characteristics of the salivary IR spectra depending on gender.

Indicator		Males, *n* = 58	Females, *n* = 42	*p*-Value
Age, years		46.4 [37.5; 53.2]	50.5 [41.5; 54.8]	0.1726
1396 cm^−1^	H	1.33 [1.15; 2.44]	3.24 [1.78; 4.03]	0.0000 *
	S	6.94 [5.92; 12.99]	18.30 [9.62; 24.80]	0.0000 *
1458 cm^−1^	H	3.21 [2.22; 4.53]	6.04 [2.95; 7.12]	0.0020 *
	S	13.7 [10.0; 22.3]	33.6 [12.3; 38.1]	0.0037 *
2853 cm^−1^	H	8.14 [6.27; 10.49]	7.85 [6.22; 11.65]	0.7171
	S	140.0 [106.0; 182.0]	142.8 [106.4; 225.5]	0.3200
2923 cm^−1^	H	13.20 [10.60; 16.02]	13.86 [10.22; 19.10]	0.4119
	S	326.9 [252.1; 395.7]	341.4 [251.0; 492.2]	0.2882
2957 cm^−1^	H	2.27 [1.81; 2.57]	2.60 [1.89; 3.00]	0.0514
	S	23.4 [18.0; 29.0]	27.2 [22.4; 48.8]	0.0026 *
2923/2957	H	5.70 [4.94; 7.18]	5.50 [4.37; 7.27]	0.2896
	S	13.5 [10.1; 20.8]	9.5 [7.2; 15.1]	0.0419 *
1458/1396	H	2.09 [1.74; 2.56]	1.72 [1.55; 1.98]	0.0001 *
	S	1.79 [1.49; 2.20]	1.50 [1.24; 1.76]	0.0005 *

Note. *—differences are statistically significant, *p* < 0.05. H is the intensity; S is the area of the corresponding absorption band.

**Table 4 diagnostics-11-01325-t004:** Characteristics of salivary IR spectra of patients with breast pathologies.

Indicator		Breast Cancer, *n* = 30	Fibroadenomas, *n* = 47	Kruskal–Wallis Criterion; *p*-Value
Age, years		56.5 [48.0; 61.5]	49.3 [40.5; 56.0]	3.028; 0.1296
1396 cm^−1^	H	1.55 [0.87; 1.96]	1.22 [0.63; 1.56]	41.04; 0.0000 *
	S	9.02 [6.13; 10.2]	5.71 [2.89; 6.89]	49.24; 0.0000 *
1458 cm^−1^	H	3.47 [2.6; 3.83]	2.33 [1.93; 3.62]	25.88; 0.0000 *
	S	18.1 [12; 20.6]	9.77 [8.16; 22]	17.98; 0.0001 *
2853 cm^−1^	H	7.85 [3.97; 9.06]	8.34 [6.97; 10.9]	4.428; 0.1093
	S	140.0 [68.2; 157.0]	151.0 [121; 206.0]	4.843; 0.0888
2923 cm^−1^	H	13 [7.76; 14.7]	14.5 [12; 18.4]	5.445; 0.0657
	S	319.5 [180; 371]	371.0 [303.0; 497.0]	6.388; 0.0410 *
2957 cm^−1^	H	2.24 [1.48; 2.5]	2.57 [2.11; 2.99]	5.078; 0.0790
	S	20.0 [14.2; 25.1]	23.5 [16.1; 29.05]	14.27; 0.0008 *
2923/2957	H	5.18 [4.54; 6.02]	5.71 [4.46; 6.54]	1.029; 0.5977
	S	15.17 [8.55; 18.87]	16.37 [11.57; 20.83]	8.422; 0.0148 *
1458/1396	H	2.45 [1.95; 3.38]	2.11 [1.73; 3.16]	23.48; 0.0000 *
	S	2.19 [1.82; 2.77]	2.21 [1.65; 3.68]	29.21; 0.0000 *

Note. The values of the Kruskal–Wallis test are given when comparing three groups: patients with breast cancer (Table 4), fibroadenomas (Table 4) and the control group (Table 3). *—differences are statistically significant, *p* < 0.05. H is the intensity; S is the area of the corresponding absorption band.

**Table 5 diagnostics-11-01325-t005:** Characteristics of salivary IR spectra of patients with prostate pathologies.

Indicator		Control (50–69 Years), *n* = 21	Prostate Cancer, *n* = 21	PINI-II, *n* = 21	Kruskal–Wallis Criterion; *p*-Value
Age, years		66.1 [62.9; 68.3]	65.0 [61.8; 69.0]	66.0 [62.0; 69.3]	2.043; 0.2889
1396 cm^−1^	H	1.30 [1.05; 2.46]	0.99 [0.90; 2.17]	1.05 [0.80; 1.18]	4.071; 0.1306
	S	6.75 [5.88; 13.43]	5.32 [4.97; 13.73]	5.34 [4.42; 6.00]	5.734; 0.0569
1458 cm^−1^	H	3.42 [3.04; 5.01]	2.40 [1.07; 5.04]	2.22 [1.28; 2.66]	9.745; 0.0077 *
	S	14.47 [12.63; 25.10]	10.69 [4.24; 26.13]	9.59 [5.17; 14.08]	8.521; 0.0141 *
2853 cm^−1^	H	8.29 [6.08; 8.74]	6.44 [5.08; 8.68]	7.15 [5.80; 8.99]	1.535; 0.4641
	S	124.2 [88.4; 145.5]	109.9 [91.7; 150.3]	122.4 [96.7; 159.2]	0.6748; 0.7136
2923 cm^−1^	H	12.64 [9.64; 13.64]	10.44 [9.46; 13.99]	11.66 [9.47; 14.28]	0.5786; 0.7488
	S	315.2 [234.4; 346.4]	279.0 [236.2; 343.7]	287.9 [225.1; 347.8]	0.4080; 0.8155
2957 cm^−1^	H	1.91 [1.59; 2.28]	1.52 [1.11; 1.84]	1.68 [1.28; 1.95]	3.443; 0.1788
	S	25.52 [18.30; 30.75]	18.75 [11.46; 27.45]	20.37 [13.60; 32.15]	1.726; 0.4219
2923/2957	H	6.34 [5.27; 7.76]	7.88 [6.09; 10.43]	7.67 [4.15; 9.87]	1.529; 0.4657
	S	12.87 [7.94; 15.85]	15.79 [10.54; 26.11]	15.26 [8.47; 22.72]	1.505; 0.4713
1458/1396	H	2.28 [1.95; 3.07]	1.77 [1.08; 2.21]	1.84 [1.36; 3.09]	6.853; 0.0325 *
	S	2.04 [1.73; 2.32]	1.63 [0.83; 2.06]	1.60 [1.11; 2.20]	6.644; 0.0361 *

Note. *—differences are statistically significant, *p* < 0.05. H is the intensity; S is the area of the corresponding absorption band.

## Data Availability

The data presented in this study are available on request from the corresponding author. The data are not publicly available because they are required for the preparation of a Ph.D. Thesis.

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
