# Peer review of "Analysis of Saliva Lipids in Breast and Prostate Cancer by IR Spectroscopy"

_diagnostics, 2021, doi:10.3390/diagnostics11081325_

Round 1

Reviewer 1 Report

The paper “Analysis of saliva lipids in breast and prostate cancer by IR spectroscopy”, by L.V. Bel’skaya et al., investigates IR spectra of lipid components extracted by saliva of patient with diagnosis of cancer, patients with non-malignant pathologies and healthy volunteers, in order to investigate the potential diagnostic value of lipid IR spectra. In fact, the authors found two markers of pathology for prostate cancer and one marker for breast cancer: these markers correspond to ratios of intensity values of specific spectral peaks related to lipids absorption. This issue is interesting because suggests the use of IR technique as complementary tools to the conventional histological images for diagnostic purpose, by considering that different changes of intensity ratios discriminate cancer and non-malignant pathologies.

However, there are several drawbacks throughout the manuscript. Such drawbacks should be properly addressed before resubmitting the manuscript to Diagnostics.

Major revisions

  1. Since “Diagnostics” is not a specialist chemometric journal, the discriminant analysis method used to classify the spectra should be briefly described in the Introduction section, to the benefit of any readers who are not expert of multivariate analysis.
  2. As for the IR spectroscopy analysis, it is necessary to specify if the background has been subtracted before calculating the areas and the intensities of the peaks.
  3. How were the peak areas calculated? By using a fitting procedure? By means of what analytical functions? If so, include a figure with a fitting example.
  4. Was the intensity of the peaks evaluated from the measured spectral value? Since the analysis was performed about the lipids extracted from the saliva samples (A spectra in Fig. 1), I wonder if the intensity of the peaks at 1396 and 1458 cm-1 is not affected by the instrumental noise, as visible in Fig. 1.
  5. Have the spectra been normalized (area? vector? SNV?). Please include such information in the Materials and Methods section. If they were not normalized, only the ratios between peaks intensity and/or area are worthy of consideration (no single values of intensity or area). So, if the spectra are not normalized, rewrite the manuscript focusing only on the peaks area ratios or intensity ratios.
  6. Figure 1: what is the attribution of bands at about 1000 cm-1 in the spectra of saliva lipids after extraction? Are they related to absorption from lipid functional groups? Why they have not been discussed?
  7. Figures 2 and 5: please insert axis titles and explain (in the text and in the figure captions) what do the circles and ellipses in Fig. 2 and 5 represent. Moreover, the meaning of positive and negative values of root 1 and root 2 should be commented.

Minor revisions

  1. Please specify the software used to perform the multivariate statistical analysis.
  2. Figure 1: insert the ticks of the horizontal axis and check the spectral positions of the vertical lines centered at the spectral position of the peaks.
  3. Figure 3: please write in the caption that the intensity values refer to the control group.
  4. Line 155: Table 3 instead of Table 2?
  5. Figures 4, 6, 7, 8 and 9: do the percentage values of intensity changes of peaks related to cancer and non-malignant pathologies with respect to the peaks of control refer to the average values? Please specify that.
  6. In my opinion, the sentences about limitations and future investigation (lines 296-303) should be moved at the end of the Conclusions section.

Author Response

See the attached file for answers to the reviewer's comments. 

Reviewer 2 Report

It is interesting phenomena of the lipid components in cancer, the lipid will be very unstable depend on the sampling to take changes also by drying and fixing treatments after the sampling as you know well. The freshness of samples will be very important to take care of the samples, which is my opinion according to a reference of 'Importance of Tissue Preparation Methods in FTIR Micro-Spectroscopical Analysis of Biological Tissues: 'Traps for New Users' reported by Zohdi, V., et al. in PLoS One, 2015, 10(2): e116491. (Show in Fig.3)

Author Response

The authors of the article are grateful to the reviewers for valuable comments that made it possible to eliminate inaccuracies and make the article better.

Reviewer 2

It is interesting phenomena of the lipid components in cancer, the lipid will be very unstable depend on the sampling to take changes also by drying and fixing treatments after the sampling as you know well. The freshness of samples will be very important to take care of the samples, which is my opinion according to a reference of 'Importance of Tissue Preparation Methods in FTIR Micro-Spectroscopical Analysis of Biological Tissues: 'Traps for New Users' reported by Zohdi, V., et al. in PLoS One, 2015, 10(2): e116491. (Show in Fig.3)

Indeed, during storage, changes can occur in samples of both biological fluids and tissues. We have indicated that the samples were not stored prior to testing in the Materials and Methods section. «Extraction and IR spectroscopic examination were performed immediately after centrifugation (without freezing and storage).»

Reviewer 3 Report

  1. In the Introduction Author should show other materials using in diagnosis of cancer by FTIR e.g. tissues (https://doi.org/10.1038/srep37333; https://doi.org/10.1016/j.saa.2017.06.021).
  2. Did Authors normalize spectra?
  3. In the Figure 1, amide I region it contains a component of the OH groups in water. A methodology (https://doi.org/10.1016/j.jpba.2017.11.074) should have been used to eliminate the water bands. Please discuss this issue.

Author Response

(The authors gave the same response as above.)

Round 2

Reviewer 1 Report

I am satisfied with the authors' responses to my comments and I agree with the changes made to the manuscript. 

I have a further comment to make: for the benefit of readers who do not know the Russian language, please briefly describe the method for the quantitative determination of lipids in saliva by IR spectroscopy (lines 47-49).

After entering such a synthetic description, the manuscript can be published in Diagnostics.

Author Response

The authors express their sincere gratitude to the reviewer for working with the manuscript, which made it much better.
We have added general information about the developed method; a detailed description of the extraction procedure is given in the Materials and Methods section. We have also added a link to the English-language article next to the article in Russian.

Reviewer 3 Report

I accept revision manuscript version.

Author Response

The authors express their sincere gratitude to the reviewer for working with the manuscript, which made it much better.